# Tumor Response on Diagnostic Imaging after Proton Beam Therapy for Hepatocellular Carcinoma

**DOI:** 10.3390/cancers16020357

**Published:** 2024-01-14

**Authors:** Hikaru Niitsu, Masashi Mizumoto, Yinuo Li, Masatoshi Nakamura, Toshiki Ishida, Takashi Iizumi, Takashi Saito, Haruko Numajiri, Hirokazu Makishima, Kei Nakai, Yoshiko Oshiro, Kazushi Maruo, Hideyuki Sakurai

**Affiliations:** 1Proton Medical Research Center, Department of Radiation Oncology, University of Tsukuba Hospital, Tsukuba 305-8576, Ibaraki, Japan; niitsu@pmrc.tsukuba.ac.jp (H.N.); lyn19960714@hotmail.com (Y.L.); nakamura@pmrc.tsukuba.ac.jp (M.N.); tishida@pmrc.tsukuba.ac.jp (T.I.); iizumi@pmrc.tsukuba.ac.jp (T.I.); saitoh@pmrc.tsukuba.ac.jp (T.S.); haruko@pmrc.tsukuba.ac.jp (H.N.); hmakishima@pmrc.tsukuba.ac.jp (H.M.); knakai@pmrc.tsukuba.ac.jp (K.N.); ooyoshiko@pmrc.tsukuba.ac.jp (Y.O.); hsakurai@pmrc.tsukuba.ac.jp (H.S.); 2Department of Radiation Oncology, Tsukuba Medical Center Hospital, Tsukuba 305-8558, Ibaraki, Japan; 3Department of Biostatistics, Institute of Medicine, University of Tsukuba, Tsukuba 305-8576, Ibaraki, Japan; maruo@md.tsukuba.ac.jp

**Keywords:** proton beam therapy, radiotherapy, diagnostic imaging, hepatocellular carcinoma, local control

## Abstract

**Simple Summary:**

Radiation therapy for hepatocellular carcinoma is known to take time to shrink tumors. In recent years, there have been an increasing number of reports on proton beam therapy, but there are few reports on how tumors shrink after proton beam therapy. In this study, it was found that tumor shrinkage after proton beam therapy also takes time in hepatocellular carcinoma as in photon radiation therapy, with PR + CR rates of 57.5% at 1 year, 76.9% at 2 years, and 85.2% at 3 years. Moreover, in many cases, the tumor still shrinks after 1 year. Therefore, it is clear that at least one year of follow-up is necessary after irradiation to confirm the shrinkage effect of proton beam therapy for hepatocellular carcinoma.

**Abstract:**

Background: Follow-up after treatment for hepatocellular carcinoma (HCC) can be mostly performed using dynamic CT or MRI, but there is no common evaluation method after radiation therapy. The purpose of this study is to examine factors involved in tumor reduction and local recurrence in patients with HCC treated with proton beam therapy (PBT) and to evaluate HCC shrinkage after PBT. Methods: Cases with only one irradiated lesion or those with two lesions irradiated simultaneously were included in this study. Pre- and post-treatment lesions were evaluated using Response Evaluation Criteria in Solid Tumors (RECIST) by measuring the largest diameter. Results: The 6-, 12-, and 24-month CR + PR rates after PBT were 33.1%, 57.5%, and 76.9%, respectively, and the reduction rates were 25.1% in the first 6 months, 23.3% at 6–12 months, and 14.5% at 13–24 months. Cases that reached CR/PR at 6 and 12 months had improved OS compared to non-CR/non-PR cases. Conclusions: It is possible that a lesion that reached SD may subsequently transition to PR; it is reasonable to monitor progress with periodic imaging evaluations even after 1 year of treatment.

## 1. Introduction

Hepatocellular carcinoma (HCC) is the most common cause of death among malignant neoplasms [1]. Surgical resection, liver transplantation, and radiofrequency ablation (RFA) are considered as curative treatments for HCC. In cases in which surgery and RFA are unsuitable, transcatheter arterial chemoembolization (TACE) is an option for patients who are not eligible for curative treatment [2,3]. Radiotherapy has historically been considered in cases for which standard treatment is not applicable, but has not been a definitive treatment because the liver is highly radiosensitive organ that is affected by low radiotherapy doses. However, recent technological developments such as intensity modulated radiotherapy (IMRT), stereotactic body radiotherapy (SBRT), and particle therapy (such as proton beam therapy (PBT) and heavy particle therapy) have increased the dose to the tumor, decreased the dose to normal tissues, and have good efficacy and safety [4,5,6,7]. Proton beams can be focused on the tumor location and delivered based on the shape of the tumor because of the Bragg peak, which makes the dose concentration in PBT useful for localized treatment of various cancers [8,9].

Follow-up after treatment for HCC can be performed using multiple imaging modalities, with dynamic CT and MRI mostly used. Solid tumors are generally evaluated using Response Evaluation Criteria in Solid Tumors (RECIST), and HCC is evaluated using modified RECIST (mRECIST) with a consideration of contrast effects. The guidelines from the Japan Society of Hepatology mentioned that no tumor regrowth or appearance of early enhancement for more than 6 months after radiation therapy is defined as local control [10]. However, there is no common evaluation method after radiotherapy. Also, there were few reports that mentioned tumor change after 6 months. It has been suggested that the contrast effect declines faster than the decrease in tumor diameter, and response rates have gradually increased [11]. The difference between CT values of the lesion and liver parenchyma in the portal phase at 1–2 years after PBT is predictive of local recurrence [12]. However, the factors related to the rate of shrinkage, such as the size of the tumor before treatment and dose fractionation, are unclear. After liver resection, local recurrence is related to tumor size, low liver function, tumor markers, and tumor differentiation, but there are few reports post-radiotherapy [13]. Some studies have described follow-up for evaluation of tumor size and contrast effect over 2–3 years, but long-term imaging changes over this period have not been examined. Therefore, the purpose of this study is to examine the factors involved in tumor reduction and local recurrence in patients with HCC treated with PBT and to evaluate HCC shrinkage after PBT.

## 2. Material and Methods

### 2.1. Patients

The subjects were patients with HCC who were treated with PBT between January 2008 and December 2018 at our center. The indications of PBT for HCC were inoperable cases (including refusal of surgery), difficult cases to treat with RFA, few residual tumors after TACE, and no liver insufficiency. Most cases were difficult to diagnose pathologically; hence, all were diagnosed with CT, MRI, and results of blood tests. Imaging was performed before and after treatment, and the images were available for evaluation in our medical records system. Post-treatment evaluations at our hospital were made by radiologists, and evaluations at other hospitals were made by radiation oncologists.

### 2.2. Proton Beam Therapy

For treatment planning, a metallic marker was inserted near the tumor under echo guidance for location identification. CT was performed in slices of 5 mm in the expiratory phase using a respiratory synchronization system (Anzai Medical Co., Tokyo, Japan). All images were transferred to a treatment planning system (VQA, Hitachi, Ltd., Tokyo, Japan). Based on pre-treatment CTs and MRIs, gross tumor volume (GTV) was contoured, and clinical tumor volume (CTV) was calculated by adding 5–10 mm to the GTV. Due to respiratory variation, an additional margin of 5 mm was added to the CTV. In addition, the multileaf collimator was enlarged and widened by a 5–10 mm margin. Proton beams were generated using the PROBEAT system (Hitachi, Ltd., Tokyo, Japan). The relative biological effect (RBE) of the proton beam was set at 1.1.

The following three protocols were used: protocol A, 66 Gray equivalents (Gy (RBE)) in 10 fractions (fr) for peripheral lesions located away from the gastrointestinal tract and porta hepatis; protocol B, 72.6 Gy (RBE) in 22 fr for hilar lesions; and protocol C, 74 Gy (RBE) in 37 fr for lesions requiring avoidance of gastrointestinal damage [6]. The dose constraints for organs at risk were <50 Gy (RBE) for the stomach and duodenum, and <60 Gy (RBE) for the colon.

### 2.3. Follow-Up Procedure

CTs or MRIs and blood tests were performed every 3 months for the first 2 years after PBT and every 6 months thereafter. For patients who found it difficult to continue to visit our hospital, CTs or MRIs were performed elsewhere. Moreover, data imported into our medical records system were analyzed.

### 2.4. Statistical Analysis

Cases in which no imaging was performed before or after PBT or for which no data were available in the medical records system were excluded. Cases with intrahepatic recurrence treated with PBT were also excluded. Cases with only one irradiated lesion or those with two lesions irradiated simultaneously were included in this study. In cases with multiple irradiated lesions, the larger lesion was analyzed. Pre- and post-treatment lesions were evaluated using RECIST by measuring the largest diameter. Blood test data were collected immediately before treatment. Overall survival (OS) and local control (LC) were determined using the Kaplan–Meier method (IBM SPSS Statistics ver. 28, IBM Co., Chicago, IL, USA). Local recurrence was defined as tumor enlargement or early enhancement on imaging. CR + PR rate (30% or more reduction of tumor length), 100% reduction (CR) rate, and 50% reduction rate were calculated with the Kaplan–Meier method. All included cases were analyzed, and cases which died or interrupted follow-up were terminated. A Cox proportional hazards model was used to estimate the hazard ratio and 95% confidence interval (CI). Changes in tumor diameter and associated factors before and after treatment were analyzed separately.

## 3. Results

A total of 755 patients with 894 HCC lesions were treated with PBT at our center from January 2008 to December 2018. Of these cases, 261 patients without pre- or post-treatment imaging evaluation and 112 patients who received multiple PBT treatments were excluded. Thus, a total of 451 cases that received PBT once for a single lesion or concurrent PBT for two lesions (with the larger lesion evaluated) were included in this study. The characteristics of these cases are summarized in Table 1.

The CR + PR rates were 33.1% at 6 months, 57.5% at 1 year, 76.9% at 2 years, and 85.2% at 3 years. The respective CR + PR were 32.7%, 62.1%, 74.8%, and 82.0% with protocol C (74 Gy (RBE) in 37 fr); 31.6%, 55.7%, 73.7%, and 81.3% with protocol B (72.6 Gy (RBE) in 22 fr); and 39.8%, 60.4%, 87.2%, and 96.8% with protocol A (66 Gy (RBE) in 10 fr). There were no significant differences among the three protocols (*p* = 0.134). The CR + PR rate of protocol is shown in Figure 1. The average rates of reduction of the tumor diameter were 25.1% (95%CI: 21.6–28.6) in the first 6 months, 23.3% (95%CI: 19.7–26.8) at 6–12 months, 14.5% (95%CI: 10.4–18.5) at 13–24 months, and 3.1% (95%CI: −0.1–6.4) at 25–36 months.

Treatment effects were predicted from pre-treatment diagnostic images. Times to 50% or 100% reduction of the long diameter of the tumor are shown in Figure 2 and Figure 3. Including patients lost to follow-up and those who died before 50% or 100% reduction, the 1-, 2- and, 3-year 100% and 50% reduction rates were 21.1%, 34.2, and 42.2%, and 43.1, 62.8, and 68.5%, respectively. A Fine–Gray regression model indicated that pre-treatment tumor size was significantly associated with the time to 50% or 100% reduction (Table 2).

The median follow-up time was 37.3 (95% CI: 33.9–43.4) months. Overall survival (OS) was 82.2% (95% CI: 78.3–85.5%) at 12 months, 69.9% (62.2–71.3%) at 24 months, and 51.9% (46.7–56.8%) at 36 months (Figure 4). Median survival time (MST) was 46.7 (95% CI 37.5–70.4) months with protocol A (66 Gy (RBE) in 10 fr), 35.4 (30.6–39.7) months with protocol B (72.6 Gy (RBE) in 22 fr), and 34.7 (22.8–47.6) months with protocol C (74 Gy (RBE) in 22 fr). Regarding the relationship between OS and tumor reduction, MST was 67.2 vs. 36.5 months in CR vs. non-CR cases at 6 months (*p* = 0.05), 47.5 vs. 32.9 months in PR vs. non-PR cases at 6 months (*p* = 0.001), 79.0 vs. 34.9 months in CR vs. non-CR cases at 12 months (*p* = 0.001), and 60.5 vs. 24.7 months in PR vs. non-PR cases at 12 months (*p* = 0.001). Thus, there were significant differences for all of these comparisons. Overall survival curves by treatment effect are shown in Figure 5 and Figure 6.

Local control (LC) rates were 94.5% (95%CI 91.8–96.4%) at 1 year, 90.2% (86.5–92.9%) at 2 years, and 86.3% (81.7–89.8%) at 3 years. The 3-year LC rates by protocol were 90.7% (78.7–96.1%) with protocol A, 85.9% (79.8–90.3%) with protocol B, and 83.7% (71.4–91.0%) with protocol C (Figure 4). There were no significant differences in dose fraction among the protocols (*p* = 0.129), but there was a trend for improved LC with a higher dose per fraction in protocol A compared to the other two protocols. There was also a significant difference in tumor size cutoff 4 cm (*p* = 0.011). The 3-year LC rates were 90.9% vs. 85.7% in CR vs. non-CR cases at 6 months (*p* = 0.462), 86.8% vs. 86.1% in PR vs. non-PR cases at 6 months (*p* = 0.751), 91.2% vs. 85.2% in CR vs. non-CR cases at 12 months (*p* = 0.257), and 87.3% vs. 86.7% in PR vs. non-PR cases at 12 months (*p* = 0.424). There was no significant difference for any of these comparisons. LC rates by treatment effect are shown in Figure 7 and Figure 8.

The relationships between the presence or absence of enhancement in the arterial phase and treatment results were also evaluated. For OS, there was no significant difference at 6 months (*p* = 0.147), but there were significant differences at 12 and 24 months (both *p* = 0.001). For LC, there were no significant differences at 6 and 12 months (*p* = 0.164, *p* = 0.197), but there was a significant difference at 24 months (*p* = 0.041).

## 4. Discussion

Previous studies of HCC have found 3-year LC and OS rates of 91–96% and 67–73% with SBRT [2,3,4,14,15], and 79.9–90% and 56–77.8% with PBT [5,6,7,16,17,18]. Thus, our 3-year LC and OS rates of 86.3% and 51.9% are similar to these reports. Regarding changes in size and contrast enhancement after SBRT, HCC may show tumor necrosis with no tumor size reduction, and no correlation between necrosis and size, suggesting that ischemia reduced the capacity of the immune system to absorb necrotic tissue [19]. Price et al. found reductions in the tumor diameter of 37% at 6 months and 55% at 12 months after SBRT [11]. Brook et al. reported tumor volume reductions of 24.5%/mo in the first 4 months after SBRT, 9.8%/mo at 4–9 months, and 2.7%/mo after 10 months [20]. In this study, the reduction rates were 25.1% (95%CI: 21.6–28.6) in the first 6 months, 23.3% (95%CI: 19.7–26.8) at 6–12 months, 14.5% (95%CI: 10.4–18.5) at 13–24 months, and 3.1% (95%CI: −0.1–6.4) at 25–36 months. These results are similar to previous studies showing the largest reduction rate in the first 6 months and a gradual decrease thereafter. At guidelines from the Japan Society of Hepatology, there is no common evaluation method after 6 months after radiation therapy [10]. However, a few of our cases showed tumor shrinkage after more than 1 year, although in most cases the lesion shrank significantly up to 1 year after treatment and the size was maintained thereafter. Since it is possible that a lesion that reached SD may subsequently transition to PR, it is reasonable to monitor progress with periodic imaging evaluations even after 1 year of treatment and additional treatment may not be necessary in the cases which have no progression.

Changes in size and contrast enhancement after PBT have only been analyzed in a few studies. Kim et al. reported a CR rate of 60% at 6 months and 89.4% at 12 months after PBT, with a median time to achieve CR of 6.3 months [21]. On the other hand, Takahashi et al. found no significant change in the tumor diameter between local recurrence and control groups [22]. In this study, the CR+PR rates were 33.1% at 6 months, 57.5% at 1 year, 76.9% at 2 years, and 85.2% at 3 years, and the median time to achieve CR+PR was 9.43 months. This result is poor compared to previous studies, which may be due to the larger tumor size (median tumor diameter 3.9 cm vs. 1.5 cm in previous reports). HCC was found to show tumor necrosis with no tumor size reduction, and if the primary lesion is larger, it may be difficult to obtain CR.

Three protocols of PBT for HCC used previously showed no significant local control differences and there were no prognostic factors other than three protocols such as sex, age, PS, liver function, hepatitis virus infection, prior treatment, number of tumors, tumor volume, and normal liver volume [6]. In this study, the 3-year LC rates were 83.7% at 74 Gy (RBE) in 37 fr, 85.9% at 72.6 Gy (RBE) in 22 fr, and 90.7% at 66 Gy (RBE) in 10 fr, with no significant difference due to dose fractionation (*p* = 0.129). However, the 3-year CR+PR rates were 82.0%, 81.3%, and 96.8% for the respective protocols, with an improved rate using the protocol of 66 Gy (RBE) in 10 fr. Despite the absence of significance in the LC rate, the difference in the CR + PR rate suggests that the size reduction rate differs depending on the dose fractionation. The protocol with the maximum BED of 66 Gy (RBE) in 10 fr tended to have the fastest reduction rate, but the number of cases for each protocol was limited, and more accumulation of cases is needed. The final LC rate was similar regardless of the shrinkage rate, which suggests that it is important to monitor the progress of shrinkage over time.

We also examined the association between the rate of tumor shrinkage and treatment outcomes (OS and LC). Cases that reached CR/PR at 6 and 12 months had significantly better OS compared to non-CR/non-PR cases. Lin et al. also found that HCC treated with PBT reached had better OS in CR cases at 3 months in MRI follow-up [23]. Intrahepatic recurrence outside the irradiated field is a common cause of death after PBT for HCC; thus, non-CR cases may be at high risk for this recurrence. In contrast, there was no correlation between LC and early tumor shrinkage. After radiotherapy for HCC, the tumor does not necessarily shrink; hence, the use of RECIST for common tumor evaluation is difficult to employ. The achievement of maximum shrinkage takes about 6 months, and some cases continue to show lesion shrinkage after 6 months. Therefore, local recurrence is defined as tumor regrowth and the appearance of contrast effects. Factors for local recurrence after resection include tumor size, low liver function, and tumor markers [24]. However, no significant factors have been reported after PBT [21,25]. In this study, there were no significant differences for liver function and tumor markers, but there was a difference for a tumor size cutoff of 4 cm.

After SBRT, treated tumors show response, progression, or stability, and there is a decrease in contrast enhancement and a reduction in size over several months for tumors with a response [2,11,24,26,27]. The decrease in contrast enhancement effect may be earlier than the decrease in tumor diameter at 6–12 months after SBRT [11,28], and there is no change in size within 3 months after SBRT [11,29]. Brook et al. found a decrease in contrast enhancement in CT at 15–45 days after irradiation, with persistence of this effect thereafter [20]. On the other hand, early enhancement and washout may remain at 12 months after treatment for a tumor with a pathological response, and residual enhancement does not quite tally with the residual tumor [30]. Sanuki et al. found that a few cases had early enhancement for more than 2 years among CR cases using RECIST [12].

Regarding outcomes, Kim et al. found that tumors with reduced enhancement at 1 month after SBRT have good outcomes [13]. Takahashi et al. reported that a difference in CT values between post-treatment lesions and liver parenchyma in the portal phase at 1–2 years after PBT is a predictor of local recurrence [22]. Lin et al. found a median time to resolution of early enhancement on MRI of 5 months after PBT [23]. In this study, the presence of early enhancement at 6 months after PBT was not related to MST (presence vs. absence: 38.4 vs. 41.9 months, *p* = 0.147), but was significant at 12 and 24 months. Early enhancement was not related to LC at 6 and 12 months, but was significant at 24 months. Some reports for SBRT indicate that enhancement in the arterial phase disappears 6–12 months after treatment and only remains in a few cases after 24 months, with early disappearance of enhancement leading to good results. However, in this study, the absence of enhancement in the arterial phase at 6 months had no effect on outcomes, including for large tumors and cases with no tumor size decrease. On the other hand, at more than 1 year after PBT, there were differences in OS, which may reflect local recurrence and intra- or extrahepatic recurrence. For LC, there was a difference at only 24 months, which is similar to a report showing that a difference in CT values in the portal phase at 1–2 years after PBT can predict local recurrence [22]. Thus, in post-treatment changes, a residual contrast effect does not necessarily indicate recurrence, and this should be evaluated in comparison with tumor shrinkage and tumor markers.

According to some guidelines, standard locoregional treatment for HCC is surgery or radiofrequency ablation. In cases which are too complicated for locoregional treatment, radiation therapy is an option [10,31,32,33,34,35]. In particular, PBT could make the dose concentration threshold for a tumor and have a lower dose for a normal liver than photon therapy [36]. However, many reports comparing PBT and other treatments were retrospective studies [17,37,38,39,40], and there was a report which investigated operable or ablation-treatable HCC [41]. Gradually, randomized comparative trials between PBT and other treatments are being reported [42,43]. Currently, some trials comparing PBT with locoregional treatment are underway, such as JCOG (Japan Clinical Oncology Group) 1315c, NCT02640924. Depending on the results of these trails, PBT has the possibility of becoming the primary treatment, replacing the current locoregional treatment. PBT is a promising initial treatment for conditions for which surgery or RFA would normally be indicated, but which are difficult to treat due to age or complications, or for conditions such as PVTT—for which surgery or RFA is difficult.

There are some limitations that should be noted. First, this study had a retrospective observational design and included some cases in which follow-up was terminated early. Second, this study used three protocols, but protocol B was most common, which may have introduced bias. Third, cases with a single lesion and the larger lesion in cases with multiple lesions were included; however, selection bias is still a concern. Fourth, the possibility that the obtained data from sources other than imaging may be involved with tumor reduction could not be ruled out. Thus, it is necessary to examine how the rate of reduction and outcomes are affected by the treatment of multiple lesions at the same time or at different times.

## 5. Conclusions

The 6-, 12-, and 24-month CR+PR rates after PBT were 33.1%, 57.5%, and 76.9%, respectively. The reduction rates were 25.1% in the first 6 months, 23.3% at 6–12 months, and 14.5% at 13–24 months. The cases that reached CR/PR at 6 and 12 months had improved OS compared to non-CR/non-PR cases.

## Figures and Tables

**Figure 1 cancers-16-00357-f001:**
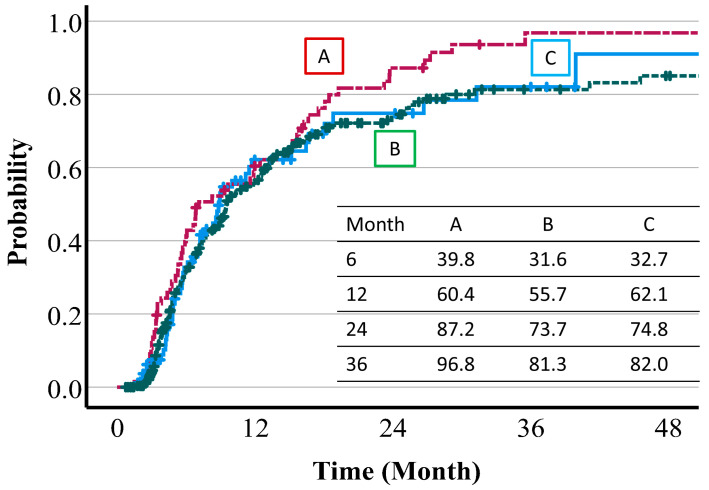
CR + PR rate by protocol. Protocol A is 66.0 Gy (RBE) in 10 fr, protocol B is 72.6 Gy (RBE) in 22 fr, and protocol C is 74.0 Gy (RBE) in 37 fr.

**Figure 2 cancers-16-00357-f002:**
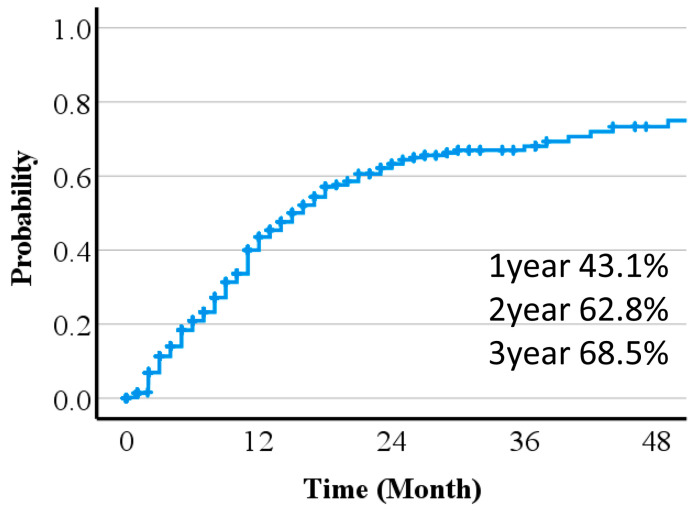
Time to 50% reduction of the long diameter of the tumor.

**Figure 3 cancers-16-00357-f003:**
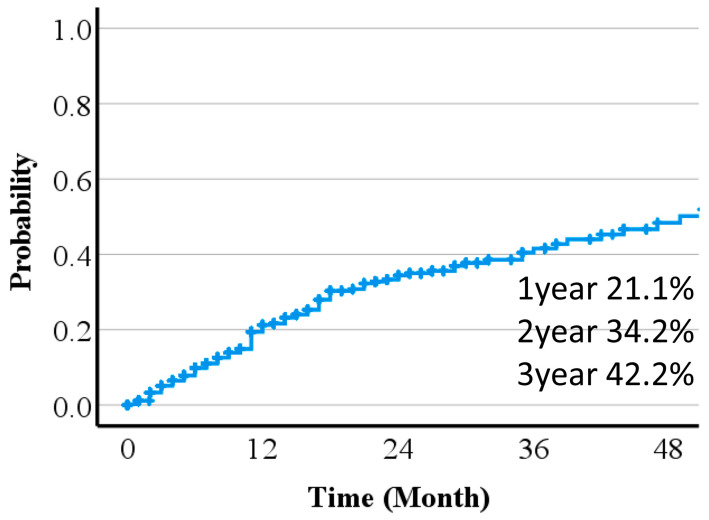
Time to 100% reduction of the long diameter of the tumor.

**Figure 4 cancers-16-00357-f004:**
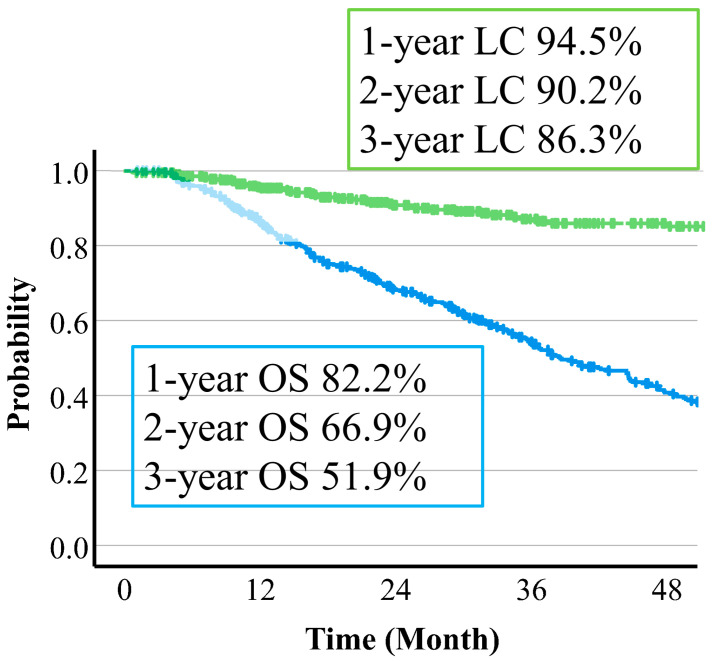
Overall survival rate and local control rate for all patients.

**Figure 5 cancers-16-00357-f005:**
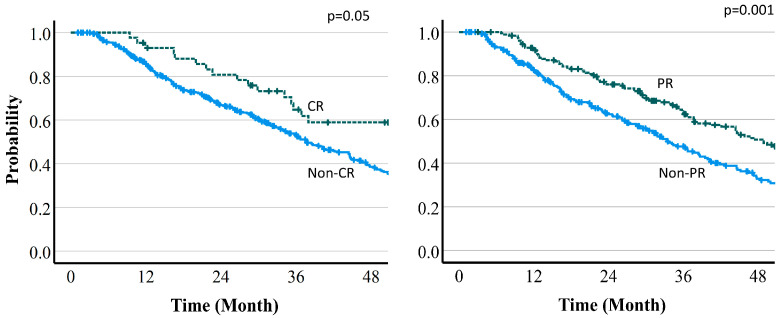
Overall survival curves by treatment effect (6-month PR or CR vs. non-PR or non-CR).

**Figure 6 cancers-16-00357-f006:**
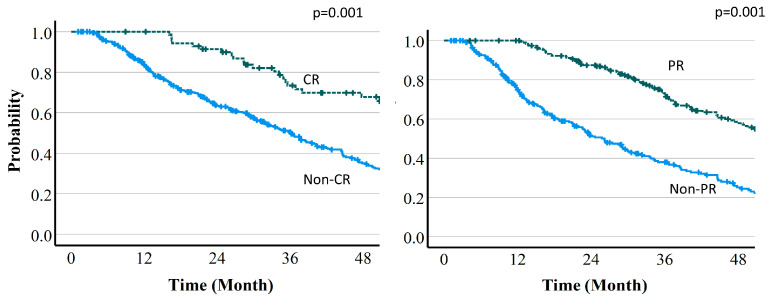
Overall survival curves by treatment effect (12-month PR or CR vs. non-PR or non-CR).

**Figure 7 cancers-16-00357-f007:**
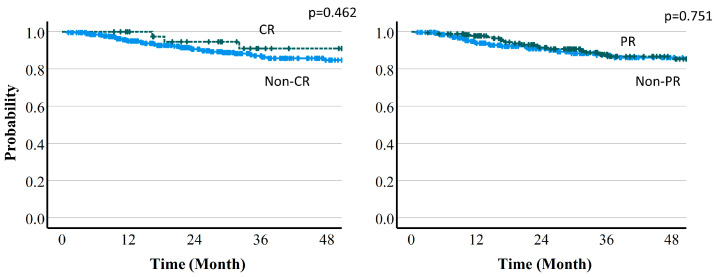
Local control rate by treatment effect (6-month PR or CR vs. non-PR or non-CR).

**Figure 8 cancers-16-00357-f008:**
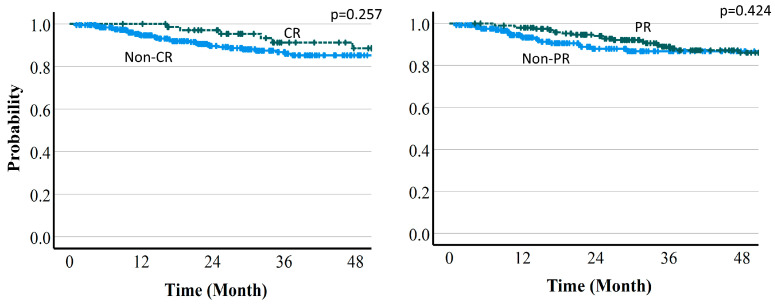
Local control rate by treatment effect (12-month PR or CR vs. non-PR or non-CR).

**Table 1 cancers-16-00357-t001:** Patients characteristics.

Characteristics	Number	%
Age (years)	21–91	72 (median)
Gender		
Male	339	75.2
Female	112	24.8
ECOG performance status		
0	244	54.1
1	182	40.3
2	24	5.3
3	3	0.7
History of hepatitis		
Non-B, non-C	204	45.2
HCV	177	39.2
HBV	70	15.5
Child–Pugh class		
A	321	71.2
B	123	27.3
C	7	1.5
Tumor location		
Peripheral	66	14.6
Hepatic portal	301	66.7
Gastrointestinal proximity	84	18.6
Tumor size (mm)	8–160	39 (median)
<30	162	35.9
30–49	149	33.0
50–99	117	25.9
≥100	23	5.1
Portal vein tumor thrombus		
Vp 0–2	400	88.7
Vp 3–4	51	11.3
Surgical indication		
Operable	68	15.0
Inoperable	383	84.9
Prior treatment		
Yes	205	45.5
No	246	54.5
Prior radiotherapy		
Yes	15	3.3
No	436	96.7
Clinical stage		
I	173	38.4
II	126	27.9
III	87	19.3
IV	65	14.4

ECOG, Eastern Cooperative Oncology Group.

**Table 2 cancers-16-00357-t002:** Fine–Gray regression model for factors associated with time to 50% or 100% reduction in the long diameter of the tumor.

Factors	HR	*p* Value
50% Reduction		
Sex	0.94 (0.68–1.30)	0.69
Age	1.00 (0.99–1.01)	0.96
Protocol A (control C)	1.10 (0.71–1.71)	0.67
Protocol B (control C)	0.79 (0.54–1.14)	0.21
Tumor size	0.89 (0.84–0.96)	<0.01
100% Reduction		
Sex	0.81 (0.53–1.24)	0.336
Age	1.00 (0.98–1.02)	0.750
Protocol A (control: C)	1.66 (0.89–3.09)	0.113
Protocol B (control: C)	1.34 (0.75–2.39)	0.328
Tumor size	0.54 (0.46–0.63)	<0.01

## Data Availability

All data used in this study are available from the corresponding author upon reasonable request.

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
