# Peer review of "Tumor Response on Diagnostic Imaging after Proton Beam Therapy for Hepatocellular Carcinoma"

_cancers, 2024, doi:10.3390/cancers16020357_

Round 1

Reviewer 1 Report

Comments and Suggestions for Authors

The manuscript by Hikaru Niitsu et al. presents intriguing data on the rates of Complete Response (CR) and Partial Response (PR) following Proton Beam Therapy (PBT) at various time intervals in hepatocellular carcinoma (HCC) patients. While the findings and conclusions are noteworthy, several aspects of the study could be enhanced to improve its overall quality and clarity.

1. The study compares CR+PR rates across three different protocols, but these protocols involve numerous variables beyond just fractionation differences, such as lesion locations. The impact of these variables on CR+PR rates is not thoroughly examined. It would be beneficial for the authors to address how these differing variables, particularly lesion locations, might influence the CR+PR rates.

2. The specific patient populations used to calculate CR+PR rates at each time point are not explicitly defined. It is unclear whether all patients were included in these calculations or only those who survived. Clarification on this point would enhance the understanding of the study's scope and the applicability of its findings.

 3. The study presents combined data for CR and PR. Separating these two outcomes could provide a more nuanced understanding of the efficacy of PBT in treating HCC. This separation would allow for a clearer assessment of the proportion of patients achieving complete eradication of the tumor versus those experiencing a significant reduction in tumor size.

 4. The study suggests the need for periodic imaging evaluations even after one year of treatment. However, it is essential to contextualize this recommendation within existing practices. If periodic imaging post-one year is already a standard practice, the study's conclusion in this regard may not provide new insights. Elaboration on how the study's recommendations align with or differ from current practices would be beneficial.

Author Response

The manuscript by Hikaru Niitsu et al. presents intriguing data on the rates of Complete Response (CR) and Partial Response (PR) following Proton Beam Therapy (PBT) at various time intervals in hepatocellular carcinoma (HCC) patients. While the findings and conclusions are noteworthy, several aspects of the study could be enhanced to improve its overall quality and clarity.

1. The study compares CR+PR rates across three different protocols, but these protocols involve numerous variables beyond just fractionation differences, such as lesion locations. The impact of these variables on CR+PR rates is not thoroughly examined. It would be beneficial for the authors to address how these differing variables, particularly lesion locations, might influence the CR+PR rates.

Reply: Thank you for the review. As you pointed out, we cannot rule out the possibility that the presence or absence of hepatitis and liver function etc, which were not used as analysis factors in this study, may have played a role in the results. Previous reports from our institution have shown that these factors are not related to local control or survival, so they were not included in present analysis. I am afraid to say that additional analysis is difficult because data other than those obtained from imaging findings, such as liver function and the presence or absence of hepatitis, were not obtained. Limitations of this study and previous reports from our institution were added in the Discussion and Conclusion (L217-230, 291-293).

2. The specific patient populations used to calculate CR+PR rates at each time point are not explicitly defined. It is unclear whether all patients were included in these calculations or only those who survived. Clarification on this point would enhance the understanding of the study's scope and the applicability of its findings.

Reply: Thank you for pointing that out. CR+PR rate, 100% reduction rate (CR rate) and 50% reduction rate were calculated by the Kaplan-Meier method, all included cases were analyzed and cases which died or interrupted follow-up were terminated. We added this explanation with methods. (L110-112)

3. The study presents combined data for CR and PR. Separating these two outcomes could provide a more nuanced understanding of the efficacy of PBT in treating HCC. This separation would allow for a clearer assessment of the proportion of patients achieving complete eradication of the tumor versus those experiencing a significant reduction in tumor size.

Reply: Thank you for pointing that out. This analysis was performed by defining a reduction of 50% or more as a reduction above PR (CR+PR: 30% or more reduction of tumor length) and a reduction of 100% as CR. We added sentence to clarify this distinction. (L110-112)

4. The study suggests the need for periodic imaging evaluations even after one year of treatment. However, it is essential to contextualize this recommendation within existing practices. If periodic imaging post-one year is already a standard practice, the study's conclusion in this regard may not provide new insights. Elaboration on how the study's recommendations align with or differ from current practices would be beneficial.

Reply: Thank you for pointing that out. The guidelines in Japan mentioned that no tumor regrowth or appearance of early enhancement for more than 6 months after radiation therapy is defined as local control, but there is no common evaluated method. Also, there are few reports that mentioned efficacy after 6 months. The rate of tumor shrinkage had slowed gradually, but there were some cases which had a trend of shrinkage after 1 year. To clarify the merit of this manuscript we added sentence in the Introduction and Discussion. (L53-57,199-206)

Reviewer 2 Report

Comments and Suggestions for Authors

1. Line 125; The average rates of reduction of the tumor..... and 0.68%/mo at 25-36 months. The rates of percentages were accumulated data or by a single period. If by single period, please take a consideration to change into accumulated number.

2. Do want to make an opinion if PBT used for the primary treatment method for HCC in the section of discussion?

Author Response

1. Line 125; The average rates of reduction of the tumor..... and 0.68%/mo at 25-36 months. The rates of percentages were accumulated data or by a single period. If by single period, please take a consideration to change into accumulated number.

Reply: Thank you for pointing that out. We evaluated the average rates of tumor reduction by single period. We changed into accumulated number in this presentation. (L129-132, 195-198)

2.Do want to make an opinion if PBT used for the primary treatment method for HCC in the section of discussion?

Reply: As you suggested we added the explanation of the possibility of primary treatment for PBT in the discussion. (L274-286)

Round 2

Reviewer 1 Report

Comments and Suggestions for Authors

Thanks for addressing my comments. I have no further questions.